# Fungal Species Causing Maize Leaf Blight in Different Agro-Ecologies in India

**DOI:** 10.3390/pathogens10121621

**Published:** 2021-12-14

**Authors:** Vimla Singh, Dilip K. Lakshman, Daniel P. Roberts, Adnan Ismaiel, Alok Abhishek, Shrvan Kumar, Karambir S. Hooda

**Affiliations:** 1Department of Botany and Plant Physiology, Chaudhary Charan Singh Haryana Agricultural University Regional Research Station, Karnal 132001, India; 2Sustainable Agricultural Systems Laboratory, USDA-ARS, Beltsville, MD 20705, USA; dan.roberts@usda.gov (D.P.R.); Ed.Ismaiel@usda.gov (A.I.); 3ICAR-Indian Institute of Maize Research (Delhi Unit), Pusa Campus, New Delhi 110012, India; alokproteome@gmail.com; 4Department of Mycology and Plant Pathology, Rajiv Gandhi South Campus, Banaras Hindu University, Mirzapur 231001, India; shrvank@gmail.com; 5Germplasm Evaluation Division, National Bureau of Plant Genetic Resources, New Delhi 110012, India; ks.hooda@icar.gov.in

**Keywords:** leaf spot, foliar blight, foliar disease, fungal plant pathogen, pathogen identification, cultural, morpho-molecular variability

## Abstract

Foliar diseases of maize cause severe economic losses in India and around the world. The increasing severity of maize leaf blight (MLB) over the past ten years necessitates rigorous identification and characterization of MLB-causing pathogens from different maize production zones to ensure the success of resistance breeding programs and the selection of appropriate disease management strategies. Although *Bipolaris maydis* is the primary pathogen causing MLB in India, other related genera such as *Curvularia*, *Drechslera*, and *Exserohilum*, and a taxonomically distant genus, *Alternaria*, are known to infect maize in other countries. To investigate the diversity of pathogens associated with MLB in India, 350 symptomatic leaf samples were collected between 2016 and 2018, from 20 MLB hotspots in nine states representing six ecological zones where maize is grown in India. Twenty representative fungal isolates causing MLB symptoms were characterized based on cultural, pathogenic, and molecular variability. Internal Transcribed Spacer (ITS) and glyceraldehyde-3-phosphate dehydrogenase (*GADPH*) gene sequence-based phylogenies showed that the majority of isolates (13/20) were *Bipolaris maydis*. There were also two *Curvularia papendorfii* isolates, and one isolate each of *Bipolaris zeicola*, *Curvularia siddiquii*, *Curvularia sporobolicola*, an unknown *Curvularia* sp. isolate phylogenetically close to *C. graminicola*, and an *Alternaria* sp. isolate. The *B. zeicola*, the aforesaid four *Curvularia* species, and the *Alternaria* sp. are the first reports of these fungi causing MLB in India. Pathogenicity tests on maize plants showed that isolates identified as *Curvularia* spp. and *Alternaria* sp. generally caused more severe MLB symptoms than those identified as *Bipolaris* spp. The diversity of fungi causing MLB, types of lesions, and variation in disease severity by different isolates described in this study provide baseline information for further investigations on MLB disease distribution, diagnosis, and management in India.

## 1. Introduction

Maize (*Zea mays* L.) is an important cereal crop in India and ranks third in production after wheat and rice [1]. Maize has wide adaptability and is gaining popularity as evidenced by its rising production and productivity [2]. Most of the produce in India is consumed as food for humans, fodder for animals, and feed for poultry, apart from applications as industrial raw materials. Among the 35 diseases affecting crop health, *viz*. seed rots and seedling blights, root and stalk rots, foliar diseases, and ear rots, maize leaf blight (MLB) caused by *Bipolaris maydis* [(Nisikado& Miyake) Shoem] is one of the major diseases of maize [1,2]. This disease has been detected in almost all maize growing areas of India [3] and is a constraint on crop improvement programs. In India, MLB occurs in the states of Punjab, Haryana, Delhi, Uttar Pradesh, Bihar, Madhya Pradesh, Gujarat, Jammu and Kashmir, Sikkim, Meghalaya, Rajasthan, Andhra Pradesh, and Maharashtra [3].

*Bipolaris* and related genera such as *Curvularia*, *Dreschslera*, and *Exserohilum* are ascomyceteous fungi known to infect maize [4], and belong to the Class Dothideomycetes, Order Pleosporales, and Family Pleosporaceae. Although the name *Cochliobolus* (1934) has been in the literature longer than the name *Bipolaris* (1959), it is not frequently used in disease reports, and *Bipolaris* is widely applied in taxonomy [5]. The association of the genus *Bipolaris* with plants from the family *Poaceae* is very common; however, species of *Bipolaris* have been reported to infect 60 other host genera, either as saprophytes or phytopathogens [6,7]. The spread of phytopathogenic species of the genus *Bipolaris* may have occurred due to international trade [8].

Based on molecular phylogenetic analysis with the ITS1-5.8S-ITS2 region of rDNA and partial sequence of the *GAPDH* (glyceraldehyde-3-phosphate dehydrogenase) gene [9], *Drechslera*, *Bipolaris*, and *Curvularia* are distinct genera even though they share many morphological similarities [4,9,10]. In addition, *Bipolaris* and *Curvularia* both have sexual morphs in *Cochliobolus* [11]. In molecular analyses of ITS and *GAPDH* gene data, two major clades, *Cochliobolus* Group-1 and Group-2 were clustered [12]. Similar results were obtained from a combined analysis of ITS, *GAPDH*, *TEF* (translation elongation factor-1 alpha gene), and LSU (28S nrRNA gene) sequence data [13]. Group-1 included the genus *Bipolaris* (type species *B. maydis*), and Group-2 included the genus *Curvularia*, (type species *C. lunata*).

The fungus *B. maydis* ((Teleomorph: *Cochliobolus heterostrophus* (Drechsler) Drechsler) exists as four different races infecting maize across the world, *viz*. race ‘T’, ‘O’ ‘C’ ‘S’ [14]. Race ‘T’ of *B. maydis* is highly virulent and was reported to cause the devastating ‘Southern corn leaf blight’ epidemic in the USA during the 1970s, resulting in huge losses due to the extremely susceptible response of Texas Male Sterile maize lines [6,15,16,17,18]. Although race ‘T’ has been reported on seeds of *Phaseolus mungo*, *Vigna sinensis*, and *Paspalum scrobiculatum* from India [19], *B. maydis* race ‘O’ is the prevalent pathogen of maize in India and elsewhere in the world [4,20,21]. *Drechslera maydis* (synonym of *B. maydis*) on maize was reported for the first time in Punjab [22]. The occurrence of a pathotype resembling race ‘T’ in India on maize hybrids 2310 and 2420 from Ludhiana was reported for the first time in 1978 [23]. Race C is the most virulent in maize cytoplasmic male sterile line C (c-cms), currently reported only in China [20,24].

For over a decade, the severity of MLB on maize in the Indian subcontinent has been increasing and the disease has spread to areas where it was not previously reported. A remarkable variability in the range of symptoms has also been recorded. Due to this variability in symptoms, it is very important to document the species of *Bipolaris* and other associated genera infecting maize, and to precisely characterize the infecting species complex. The rising incidence of *Curvularia* sp. on maize is also a concern for crop improvement programs [25]. The objectives of this study were (1) to examine the diversity of *Bipolaris* species and associated fungal genera causing MLB across different agro-ecological zones of India, (2) to provide correct morphological descriptions and identification of the species/isolates for reference in order to device identification and management strategies, and (3) to assess the virulence spectrum of common MLB pathogenic species. Knowledge gained from this study will provide benchmark information necessary to accurately identify MLB pathogens, breed for resistant cultivars, and improve disease management practices in India.

## 2. Results

### 2.1. Survey, Collection, and Maintenance of Fungal Isolates

A total of 350 symptomatic maize leaf samples showing maize leaf blight (MLB)-like symptoms were collected from 20 hotspots representing six agro-ecological zones under maize production during survey and surveillance visits during the Kharif seasons (June–October) in 2016–2017 and 2017–2018 (Appendix A). The six hotspots, viz. Godhra, Anand, Sukheri, Lakhawali, Mandya, and Mysore showed variability in morphological and pathogenic profiles. However, in view of the sample size and constraints of the resources available for this study, one candidate isolate was selected from each of the 20 hotspots, *viz*. Pantnagar, Bajaura, Kangra, Patelnagar, Ludhiana, Pichola, Banswara, Kota, Chittorgarh, Dungarpur, Amberi, Karnal, Dholi, Samastipur, Godhra, Anand, Sukheri, Lakhawali, Mandya, and Mysore for phylogenetic analysis (Table 1). Isolates were examined for typical characteristics of *B. maydis* which is the prevalent species infecting maize in India, as reported by earlier workers [23,26,27,28]. Variability in colony growth (slow, medium, or fast), color (light grey, dark grey, light black, or black), texture (rough raised, rough appressed, smooth raised, smooth appressed with or without zonation, and regular or irregular margins), conidial dimensions (7.6 µm to 19.3 µm × 3.1 µm to 6.1 µm), pathogenicity (mild to high) and disease severity (30 to 90%) revealed that out of 20 hotspots surveyed, fungal isolates from 14 hotspots (Table 2) showed typical morphological and pathogenic characteristics of *Cochliobolus heterostrophus* (Drechs.) Drechs. (anamorph = *Bipolaris maydis* (Nisi-kado) Shoemaker; synonym = *Helminthosporium maydis* Nisikado).

### 2.2. Phylogenetic Analyses

The final ITS alignment file had thirty taxa and 483 characters of which 84 were parsimony informative representing 17% of the total characters. The *GADPH* file had 23 taxa and 512 characters of which 118 were parsimony informative representing 23% of the total characters. For both files, reference sequences representing mostly sequences of ex-type cultures of *Bipolaris* or *Curvularia* species were obtained from GenBank and were included in Appendix A. The phylogenetic trees obtained with Bayseian analyses had essentially identical topology to the trees obtained by maximum likelihood (ML). Therefore, for each locus, only one ML tree was selected for this publication, but the posterior probability (PP) support for the branches from the Bayesian trees is reflected on the maximum likelihood tree branches. The maximum likelihood tree shown in Figure 1A based on *GAPDH* sequence data reveals that three taxa under study formed a highly supported subclade (A) with a sequence of ex-type culture *B. maydis*. As a matter of fact, there were 10 additional isolates with identical sequences to taxa in clade A which were not included in the analysis due to the tree size. In other words, 13 of 20 isolates were identified as *B. maydis*. The tree based on ITS sequence data (Figure 1B) was congruent with the *GAPDH* tree and revealed the same clade (A) within a large all *Bipolaris* species clade. One isolate (*BmMdKa6*) had identical sequence to the ex-type culture of *B. zeicola* and they together formed a highly supported (BS = 97%; PP = 0.98) subclade B (Figure 1A). This subclade was sister to the *B. maydis* subclade with BS support of 100% and PP of 1. Similar to subclade A, isolate *BmMdKa6* was also identified as *B. zeicola* in the ITS tree (subclade B; Figure 1B). In brief, among the 20 isolates identified morphologically and pathologically as fungi causing MLB, 13 isolates (*viz.*
*BmPhRj4*, *BmBjUa1*, *BmBsRj4*, *BmDhBh3*, *BmCgRj4*, *BmPnDl2*, *BmDnRj4*, *BmKgUa1*, *BmKrHr2*, *BmKtRj4*, *BmLdPj2*, *BmPtUa1*, and *BmMyKa6*) representing 13 hotspots were identified as *B. maydis* and one isolate (*BmMdKa6*) was identified as *B. zeicola*, based on phylogenetic trees of two loci, *GAPDH*, and ITS.

The remaining taxa in the tree were *Curvularia* sp. isolates and they formed three main supported clades C, D, and E. (Figure 1A). Clades C, D, and E had a polytomy relationship to each other and to *Bipolaris* clades B and A. (Figure 1A). Within clade E, isolates *BmAdGj5* and *BmGdGj5* and ex-type culture of *C. papendorfii* had identical sequences, and together they formed a highly supported subclade (BS = 98% and PP = 0.99 subclade). Isolate *BmLhRj4* had an identical sequence to an ex-type culture of *C. sporobolicola*, and together they formed a highly supported subclade. Isolate *BmSkRj4* formed a subclade with two isolates of *C. siddiquii*, with high support (BS = 0.99 and PP = 1). In the ITS tree, however, the isolates that fell into clade E (*BmLhRj4*, *BmGdGj5*, *BmAdGj5*, and *BmSkRj4*) formed a highly supported clade with *C. lunata* and they all had identical ITS sequences. Clearly, phylogeny based on ITS sequence data appears to be unsuccessful in resolving *Curvularia* isolates to the species level like the *GAPDH* tree. Isolate *BmSmBh3* (Figure 1A) formed a clade with the sequence of the unidentified species of *Curvularia* (Accession No. KU552166). The species *C.*
*graminicola* was shown to be the closest relative to them (Clade D, Figure 1A). Therefore, this unidentified species remained ambiguous, needs additional markers-based phylogeny for clarity, and was named *C*. *graminicola*-like fungus for this report. In the ITS tree, isolate *BmSmBh3* fell into the large *Curvularia* clade but did not form a clade with a sequence of any known species of *Curvularia*.

The isolate *BmAmRj4* did not fall into the *Bipolaris* or *Curvularia* clades but formed a clade with the ex-type culture of *Alternaria alternata* in the ITS tree (Figure 1B). The two sequences had 100% homology. However, we could not obtain the *GAPDH* sequence for this isolate, and thus it was excluded from the phylogenetic analyses based on *GAPDH*. In summary, in addition to the 14 isolates of *B. maydis* and *B. zeicola* that were causative agents for MLB, we identified two isolates of *C. papendorfii*, and one isolate each of *C. siddiquii*, *C. sporobolicola*, a *C. graminicola*-like fungus, and an *Alternaria* sp. as MLB causing fungi in India.

### 2.3. Morphological Characterization of Fungal sp. Causing Maize Leaf Blight

The 20 candidate fungal isolates maintained on Potato Dextrose Agar (PDA) plates showed striking variability in culture morphology and conidial dimensions (Table 2; Figure 2A). The length and width of conidia of different isolates varied between 7.6 µm (*BmgGdGj5*, Godhra) to 19.3 µm (*BmPnDl12*, Patel Nagar) and 6.1 µm to 3.1 µm (*BmSmBh3*, Samastipur and *BmgGdGj5*, Godhra or *BmSkRj4*, Sukher), respectively. The isolate *BmPnDl12* (Patel Nagar) had the largest sized conidia with mean length of 19.3 µm and width of 5.0 µm and *BmgGdGj5* (Godhra) had the smallest conidia, 7.6 µm × 3.1 µm. The conidial dimensions of the rest of the 18 isolates were in the order *BmAmRj4* (Amberi) *> BmLhrj5* (Lakhawali) *> BmBjUa1* (Bajaura) *> BmSkRj4* (Sukher) *> BmKrHr2* (Karnal) *> BmCgRj4* (Chittorgarh) *> BmPtUa1* (Pantnagar) *> BmDhBh3* (Dholi) *> BmLdPj2* (Ludhiana) *> BmMyKa6* (Mysore) *> BmKtRj4* (Kota) *> BmBsRj4* (Banswara) *> BmAdGj5* (Anand) *> BmMdKa6* (Mandya) *> BmDnRj4* (Dungarpur) *> BmKgUa1* (Kangra) *> BmPhRj4* (Pichola) *> BmSmBh3* (Samastipur) (Table 2).

The number of septa of various isolates ranged between one to three (*BmgGdGj5*, Godhra isolate) to five to eight (*BmBjUa1*, Bajaura isolate) (Table 2; Figure 2A). For the rest of the isolates, it was in the order *BmAmRj4* (Amberi) *= BmLhRj4* (Lakhawali) *>* BmKrHr2 (Karnal) > BmMdKa6 (Mandya) > *BmPnDl12* (Patel Nagar) *> BmBsRj4* (Banswara) *= BmLdPj2* (Ludhiana) *> BmDhBh3* (Dholi) *> BmPtUa1* (Pantnagar) *= BmMyKa6* (Mysore) *> BmCgRj4* (Chittorgarh) *> BmAdGj5* (Anand) *> BmSkRj4* (Sukher) *> BmSmBh3* (Samastipur) *> BmKtRj4* (Kota) *= BmDnRj4* (Dungarpur) *= BmPhRj4* (Pichola) (Table 2).

Various cultural characteristics like radial growth, colony texture, color, and diameter revealed that the isolates from Banswara, Chittorgarh, Patel Nagar, Dungarpur, Kangra, Karnal, Godhra, Mandya, and Mysore had fast radial growth on PDA, whereas isolates from Pichola, Bajaura, Dholi, Kota, Pantnagar, Amberi, Lakhawali, and Samastipur were medium growing and isolates from Ludhiana and Sukher were slow growing. The isolate *BmMdKa6* (Mandya) had the fastest radial growth, whereas *BmLdPj2* had the slowest growth. Colony colors were light black (*BmPhRj4* and *BCgRj4*); light grey (*BmDnRj4*, *BmMdKa6*, *BmMyKa6*); dark grey (*BmKgUA1*, *BmKrHr2*, *BmBmLdPj2*, *BmPtUa1*, *BmLhRj4*, *BmSkRj4*, *BmSmBh3*) and black (*BmBjUa1*, *BmBsRj4*, *BmDhBh3*, *BmPnDl12*, *BmKtRj4*, *BmGdGj5*, *BmAmRj4*, *BmAdGj5*) (Figure 2B).

The culture morphology of *C. papendorfii* showed a rough surface with irregular margins and no zonation, conidial dimensions 7.6 µm × 3.1 µm to 16.9 µm × 4.1 µm, and one or more septa (Table 2) which falls into Group-2 (Appendix A); whereas *B. zeicola* showed smooth appressed surface with regular margins and zonation, conidial dimensions 7.8 µm × 6.1 µm to 13.1 µm × 3.8 µm slightly larger than *C. papendorfii*, and 3 to 4 septa (Table 2) formed Group-5 (Appendix A). The *C. papendorfii* isolate from Godhra showed a rough raised surface while the Anand isolate showed a rough appressed surface with no zonation and smooth margins. The conidial size was also very small, with 1 to 2 septa, and ranged from 7.6 to 12.1 µm × 3.1 to 4.3 µm. Ecologically both *B. zeicola* and *C. papendorfii* prevailed in different parts of the plateau and hill regions which are at high altitudes with cool and moist climates close to the sea (*C. papendorfii* from the western part having sandy loam soil and *B. zeicola* from the southern part of the plateau and hill region having laterite soils). Isolates from Lakhawali, Sukher, and Anand showed variable morpho–pathogenic profiles and represented different groups being different species in the genus *Curvularia*. The conidia of the *C. sporobolicola* (Lakhawali) isolate were slightly curved with 2 to 3 septa, conidial size ranging from 17.2 µm × 3.7 µm, forming a rough colony with zonation and irregular margins. However, the *C. siddiquii* (Sukher) isolate had conidial sizes smaller than the *C. sporobolicola* isolate (i.e., 16.8 µm × 3.1 µm in size) with a slight curvature, 3 to 4 septa, smooth appressed colony, no zonation, and rough margins. The conidial morphology of *C. papendorfii* was oval or obpyriform with 1–3 septa. The *C. graminicola*–like sp. (Samastipur) isolate produced sickle-shaped conidia with 2 to 3 septa, ranging in size from 7.8 to 6.1 µm and 2.6 µm average diameter. *Alternaria* sp. was also isolated from maize in some regions from the Amberi hotspot in Rajasthan.

### 2.4. Pathogenic Variability of Fungal Isolates on Zea mays c.v. DHM 117

Koch’s postulates experiments were performed for all 20 isolates, and all were positive for causing MLB disease. The MLB symptoms appeared as small yellowish necrotic spots 3 to 4 days after inoculation of the test plants, *Zea mays* c.v. DHM 117. Necrotic spots coalesced with the progression of the disease resulting in blight symptoms. The severity of the disease differed between the isolates (Figure 2C). In general, isolates identified molecularly as *Curvularia* spp. (SN 15 to 19, Table 2) and *Alternaria* sp. (SN20, Table 2) caused more severe disease with scores of 3.3 to 4.5 and an average score of 3.91, than isolates identified as *B. maydis* (SN 1 to 13, Table 2) or *B. zeicola* (SN 14, Table 2), with scores of 1.5 to 4.3 and an average score of 2.65. The type of lesions on the leaves differed, and thus, we categorized them into Types I to VI (Table 2). The type of lesions was overlapping between the isolates identified as *Bipolaris* spp. and those identified as *Curvularia* spp. The isolates *BmPhRj4*, *BmDhBh3*, *BmKrHr2*, *BmGdGj5*, and *BmMdKa6* produced Type-I symptoms but mild and moderate to severe virulence within 48 to 72 h after inoculation. The isolates *BmBjUa1*, *BmCgRj4*, *BmDnRj4*, *BmKgUa1*, *BmLdPj2*, and *BmPtUa1* showed Type-II symptoms with 48 to 76 h incubation period and moderate to severe virulence. Type-III symptoms were shown by isolates *BmBsRj4*, *BmPnDl2*, *BmKtRj4*, and *BmLhRj4* with 48 to 96 h incubation period and mild and moderate to severe virulence. Type-IV symptoms were observed in isolates *BmAmRj4* and *BmMyKa6* with an incubation period of 72 h and severe virulence. Isolates *BmSkRj4* and *BmSmBh3* showed Type-V symptoms with a 72 to 76 h incubation period and severe virulence. Type-VI symptoms were expressed by isolate *BmAdGj5* with a 72 h incubation period and severe virulence.

The colony texture of isolates correlated with their pathogenic characteristics resulting in the categorization of the isolates into six groups (Appendix A). Group-1, represented 64 samples (18.28%) of the total population examined, and had rough colonies with no zonation and irregular margins as shown by isolates *BmPhRj4*, *BmAmRj4*, and *BmLdPj2*. The isolates of this group had mild to moderate virulence, an incubation period of 72 h on the host, and disease severity of 1.5 to 4.3 on the rating scale; the highest virulence being recorded was for *BmLdPj2*. Group-2 with 71 samples (20.28%), had rough colonies with no zonation and regular margins, as shown by isolates *BmBjUa1*, *BmCgRj4*, *BmPtUa1*, and *BmGdGj5*, had moderate virulence, an incubation period of 48 to 72 h and disease severity of 2.6 to 3.3. Group-3 with 26 samples (7.42%) had rough colonies with zonation and irregular margins, as shown by *BmPnDl12* and *BmLhRj4*, which had moderate to high virulence, an incubation period of 72 h, and disease severity of 2.7 to 3.7. Group-4 with 110 samples (31.42%), had smooth appressed colonies with no zonation and regular margins, as shown by isolates *BmBsRj4*, *BmDnRj4*, *BmKgUa1*, *BmKrHr2*, *BmKtRj4*, and *BmSmBh3*, had mild to high virulence, an incubation period of 48 to 96 h and disease severity of 1.9 to 3.6. Group-5 with 40 samples (11.42%) had smooth appressed colonies with zonation and regular margins, as shown by isolates *BmMdKa6*, *BmMyKa6*, and *BmSmBh3* had mild to high virulence, an incubation period of 72 to76 h, and disease severity of 1.7 to 4.3. Finally, Group-6 with 39 samples (11.14%) had rough appressed colonies with no zonation and regular margins and high virulence, a 72 h incubation period, and disease rating of 3.7 to 4.5.

Based on the expression of symptoms (Type-I to -VI, Figure 2C), the isolates were categorized into six groups. The morphological and pathological variations of *B. maydis* isolates were distributed among all six groups, whereas *Curvularia* isolates were distributed from Group-2 to Group-6, and *Alternaria* sp. came under Group-1 (Appendix A). *B. maydis* was the dominant pathogenic species out of the total population of MLB collected across 20 hotspots in six maize production zones being identified in 72.85% of the 350 disease samples collected, followed by *Curvularia* species being identified in 20.28% of the samples. A few instances of *B. zeicola* (3.71%), *C. papendorfii* (3.42%), and *Alternaria* sp. (3.14%) were also detected from the MLB samples. Mixed infection of *B. maydis*, *C. papendorfii*, and *Alternaria* sp. were noticed from the same leaf in Mysore (Karnataka state), Amberi (Rajasthan state), and Anand (Gujarat state) (data not shown).

Visually the symptoms of *B. maydis* could be differentiated from *B. zeicola* in having oval lesions (4 to 5 mm × 7 to 9 mm) which later elongated and coalesced into larger irregular necrotic lesions along the mid rib (9 to 12 cm), whereas *B. zeicola* produced circular to oval dot–like yellowish necrotic lesions (2 mm × 3 mm) which did not coalesce with the progression of the disease (Figure 2C). The lesions increased in circumference but remained distinct, increasing in size up to 1.2 cm × 2.5 cm. However, for both the *B. maydis* and *B. zeicola* isolates, no lesions could be observed on husks and leaf sheaths, and no wilting was observed in the diseased plants in field observations as well as in the greenhouse inoculation experiments. This may serve as a preliminary indication that the isolates of *B. maydis* observed in our survey belong to race type “O” (20). *B. maydis* showed an extensive distribution across all maize cropping zones.

The *C. papendorfii* isolates formed very minute dot-like, yellowish necrotic spots more profuse than symptoms caused by *B. zeicola* the lesions showing irregular purplish brown margins (3 mm × 5 mm) which remained discrete and showed no coalescence with progression in size of spots, whereas *C. sporobolicola* and *C. siddiquii* formed tan colored streaks on the leaves. While *C. sporobolicola* showed coalescence of streaks to form large necrotic streaks on the leaf (Figure 2A, plate q), the *C. siddiquii* isolate formed elongated mosaic-like streaks which remained discrete and did not coalesce. Symptoms of *C. graminicola* were close to *C. papendorfii*; however, the lesions were smaller than those formed by *C. papendorfii* with no purplish margins, and coalescence with the progression of the symptoms was observed. The *C. papendorfii* and *B. zeicola* isolates were weakly to moderately pathogenic on maize in the plateau and hill regions of India. Another pathogen, *Alternaria* sp. was detected to cause MLB in India. Symptoms of *Alternaria* were visible as discrete oval to irregularly elongated, yellow necrotic spots (1.3 cm × 2.7 cm to 2.5 cm × 3.7 cm), discrete and larger in size than with *B. maydis*, *C. papendorfii*, and *B. zeicola*. With disease progression, lesions coalesced up to 2.7 cm × 3.4 cm in size, but not as large irregular elongations. Symptoms of *Curvularia* sp. generally were scattered 3 to 5 mm × 6 to 7 mm lesions, visible as mosaic patterns on the leaf lamina, which coalesced to form necrotic regions on leaves.

## 3. Discussion

Maize leaf blight (MLB) is listed as a major biotic stress on maize in India, and every year monitoring visits are undertaken to survey the disease-prone areas to examine yield losses due to the disease [1,2]. Precise characterization of the species infecting the maize crop is needed for developing effective strategies to manage the economic losses. In the global climate change scenario, it is very important to examine the trends in yield losses and the severity of MLB on the crop within different cropping zones. In the current investigation, for molecular identification of fungi obtained from MLB samples, we depended primarily on the section of the *GAPDH* gene, which has been regarded as the best single marker for delineating species of the genus *Bipolaris* [29]. Additionally, we used phylogeny based on ITS to further support of the results. *B. maydis* was identified in 13 of 20 hotspots surveyed. This observation supports *B. maydis* as the dominant MLB pathogen in India [3]. However, *B. zeicola* was present in one hotspot. These two species were the only *Bioplaris* species detected in our relatively small sample size for characterizations. Our results are in agreement with the survey in China where they found that these two species accounted for about 97% of *Bipolaris* species causing diseases in maize [20]. In Yunnan, Sichuan, and Shaanxi Provinces of China, *B. zeicola* isolates were reported to produce long, narrow linear lesions [8,20,30,31]. However, the pathogenicity of the Indian isolate of *B. zeicola* is slightly different from the Chinese isolates. Isolates of *B. maydis* from different hotspots across six maize production zones of India showed mainly Type-I, -II, and -III symptoms (Table 2; Figure 2C). Therefore, elongated, necrotic lesions were the typical symptoms caused by *B. maydis*, as reported in previous studies [3,28]. Although the association of *B. zeicola* (synonym *B. carbonum*) in healthy maize seeds [32] and on healthy rice leaves [33] in India were noted earlier, the reports lacked pathogenicity data of the respective organisms. *B. zeicola* is also reported to be a pathogen of rice and maize in China [34] and a pathogen of rice in Pakistan [35].

Of the four reported races of *B. maydis viz*. O, T, C, and S [14,36], lesions of race O were tan in color with buff to brown borders. They began as small, diamond-shaped lesions and sometimes elongated within the veins to become larger and rectangular. However, race O lesions were contained within the leaves. Lesion size ranges from 2 to 6 mm × 2 to 22 mm. Lesions produced by race T were oval and larger than those produced by race O and isolates of race T commonly affected husks and leaf sheaths. Lesions caused by race C were necrotic and found to be about 5 mm long. They also tended to cause wilt [14]. While it is beyond the scope of this study, based on symptomatology we conclude that the *B. maydis* isolates observed in our survey belong to race O. Race O was also found to be the predominant pathogen of maize in India by other workers [27,37]. Studies on the virulence of *B. maydis* and *B. zeicola* isolates revealed that both are adapted to distinct ecological conditions [31,38,39]. It was reported that race 3 of *B. zeicola* with narrow linear lesions on the leaves of mature maize plants may have been a mountain ecotype, favoring high humidity and cool temperatures at high elevations for infection. However, in our study *B. zeicola* produced dot-like, spherical lesions.

In recent years variations in MLB symptoms on maize have been noticed in India. This points towards the likely establishment of other species/races also infecting maize. In substantiation, we report for the first time that along with *B. zeicola*, four *Curvularia* pathogens (*C. papendorfii*, *C. sporobolicola*, *C. siddiquii*, *C. graminicola*–like fungus), and *Alternaria* sp. are pathogens causing MLB disease in India. Earlier, a leaf spot disease of maize caused by *C. clavata* and maize leaf spot caused by *C. geniculata* were recorded in India [25,40]. The presence of *C. papendorfii* in rice soil [41] has been documented from India but the report lacked any pathogenicity data for the organism. However, the presence of four *Curvularia* species in five out of 20 molecularly identified pathogens raises concern that *Curvularia* is an emerging threat to maize in India. *C. sporobolicola* was shown to be a pathogen of the grass *Sporobolus australasicus* in Australia [42] and *C. siddiquii* a pathogen of *Pennisetum americanum* in India [43], *C. graminicola* was isolated from *Aristida ingrata* (Poaceae) in Australia [44], and a taxonomically close relative of the fungus (*Curvularia* sp. BRIP 61674) was found to be a pathogen of *Oryza* spp. in Australia [45]. Furthermore, *C. papendorfii* (synonym: *B. papendorfii*) was shown to be a pathogen of maize in China [46]. *Alternaria* species including *A. tenuissima*, *A. alternata*, and *A. burnsii* were shown to cause disease in maize [47,48]. Migration of these pathogenic fungal species to maize may have happened from adjacent sugarcane or rice fields or due to secondary inoculum developed on previous crops in the rotation schedule, which necessitates further investigations.

Previous investigations on the characterization of maize pathogens from MLB disease prone maize production zones indicated the presence of the disease caused by *B. maydis* in all the maize production zones, especially in Kharif maize [3,28]. *C. papendorfii* and *B. zeicola* were characterized by Godhra and Mandya hotspots which were in the plateau and hill region. In addition to the widely distributed *B. maydis* and minor reports of *B. zeicola*, some other *Bipolaris* species such as *B. sorokiniana* have also been reported in other countries as harmful pathogens of maize [4,49,50]. Interestingly, *B. sorokiniana* that causes wheat root rot and leaf spot was the dominant species infecting wheat in India [51]. Therefore, the chance establishment of this species on maize, particularly in traditional wheat–maize rotations is probable. *B. sorokiniana* has been reported from maize fields under wheat–maize rotation in Sichuan [20]. Here it is worth mentioning that reports of *B. sorokiniana* as a dominant species infecting wheat in India can be an emerging threat to maize because the crop cycle of spring maize has some overlap with the wheat season. Similarly, *B. sacchari* a pathogen of sugarcane in India [43], was also reported as a pathogen of maize in China [52]. Recently, a new sheath spot disease of maize caused by *Waitea circinata var*. *prodigus* has been reported from eastern India [53]. Therefore, regular monitoring in maize fields for the possible presence of new or emerging pathogens along with *B. maydis*, *B. zeicola*, *Curvularia* spp., and *Alternaria* sp. (this report) are necessary to document various fungi causing MLB in India.

Our studies on pathogenicity supported previous reports, however, severity varied for different isolates. Although isolates of *B. zeicola*, *C. papendorfii*, *C. sporobolicola*, *C. siddiquii*, and *C. graminicola*–like pathogens showed weak, moderate to severe virulence on maize, their occurrence is a concern for the Western and Southern Plateau and Hill region. Their incidence may rise in the future with changing environments. Hence, it is essential to test the virulence on maize lines and to establish the lines showing severe infection for advisories to avoid huge maize losses. We suggest a more rigorous screening of maize germplasm under simulating epiphytotic conditions to examine the pathogenicity of species of *Bipolaris*, *Curvularia*, and *Alternaria* at high altitudes and cold regimes. The race diversity of *B. maydis*, *B. zeicola*, *Curvularia* spp., and *Alternaria* sp. on maize also needs to be investigated to avoid complications of possible mixed infection.

Taken together, we explored the species diversity of fungi causing MLB in maize production zones of India based on cultural morphology and symptoms on the host. We found that *B. maydis* was the dominant species infecting maize in all geographical locations surveyed. The isolates of *B. maydis* also showed a variation in symptoms on *Zea mays* c.v. DHM 117. The *B. zeicola*, *C. papendorfii*, *C. sporobolicola*, *C. siddiquii*, *C. graminicola*–like fungus, and *Alternaria* sp. are six new species we identified to cause MLB in India. Of these *C. sporobolicola*, *C. siddiquii*, *C. graminicola*–like fungus are probably the first worldwide reported as pathogens of maize. The symptoms of the four *Carvularia* species and the *Alternaria* sp. recorded on maize in India were more severe than the dominant species *B. maydis* probably due to more rigorous crop management strategies against the target species. These species recorded in the study reported here were occasionally able to cause mixed infections in the field but were distinguished by pure culture isolations and symptom expressions on test plants. These findings may contribute greatly to the understanding of the species diversity in the maize production zones of India and aid in the diagnosis of MLB pathogens and management.

## 4. Materials and Methods

### 4.1. Collection and Maintenance of Isolates

Maize leaves showing characteristic maize leaf blight (MLB)–like symptoms (n = 350) were collected from maize fields across the different agro-climatic zones of India covering the states of Uttaranchal, Himachal Pradesh, Karnataka, Haryana, Delhi, Rajasthan, Bihar, Gujarat, and Punjab (Appendix A). Surveys were undertaken in disease prone areas to collect different fungal isolates from maize showing leaf spots and blights (Table 1). Symptomatic samples were thoroughly washed in sterile water and 1–2 mm bits of infected leaf tissue showing lesions were cut and surface sterilized using 2% sodium hypochlorite for 5 min, washed with sterile water, and blotted dry. The sterilized bits were then transferred aseptically into Petri plates containing PDA These plates were incubated at 25 ± 1 °C in a BOD Incubator (REMI Cl-10, Mumbai, India). Pure cultures of isolates were established by single spore isolation and examined under a light microscope (Olympus BX-53, Tokyo, Japan) to study characteristic features. The fungal isolates were maintained at standard storage conditions on PDA slants for further studies. From a total of 350 samples analyzed, 20 fungal isolates representing candidate isolates for each location surveyed were further examined to assess the MLB causing fungal species diversity.

### 4.2. Fungal DNA Extraction, Amplification, Sequencing and Phylogenetic Analysis

Growth plugs (10 mm diameter) from actively growing 7-day-old cultures of 20 fungal isolates (Table 1) maintained on PDA were inoculated into 100 mL of potato dextrose broth media in Erlenmeyer flasks and incubated at 25 °C in a BOD incubator, with shaking at 100 rpm. Mycelia were harvested with a sterile Whatman No. 4 filter disk and Buchner funnel attached to a vacuum flask. Then, mycelia were washed with sterile distilled water, blotted dry between layers of tissue paper, wrapped in aluminum foil, frozen in liquid nitrogen, and stored at −80 °C until needed.

Genomic DNA was extracted using the Cetyltrimethylammonium bromide (CTAB) method [54]. DNA concentrations were determined using a Pico-drop Spectrophotometer (Pico-drop Ltd., Cambridge, UK), and the concentration adjusted to 200 ng/µL. DNA solutions were stored at −80 °C until used.

PCR was performed to amplify the internal transcribed region of the rRNA gene using primers ITS1 and ITS4 [55]. The PCR mixture (50 µL) contained 10 µL of 5× PCR Go TaqFlexi buffer (Promega Corp., supplied by Pragati Biomedical, New Delhi, India), 0.2 mM dNTPs, 0.2 µM of each primer, and 1.25 units of Taq Polymerase GoTaq Flexi (Promega Corp., Pvt. Ltd., Mumbai, India). Reaction mixtures in PCR vials were placed in a thermocycler (CT-100 Bio-Rad, Gurugram, India). The program used for amplification consisted of an initial 2 min denaturation at 94 °C, followed by 35 cycles of 1 min denaturation at 94 °C, 1 min annealing at 55 °C, and 1 min extension at 72 °C. The program was concluded with a 10 min extension at 72 °C. PCR amplification was checked by gel electrophoresis on 1% agarose gels stained with ethidium bromide. PCR products of 439 to 587 bp were excised from the gel and cleaned using the Wizard^®^ SV Gel and PCR Clean–Up System (Promega Corp., Madison, WI, USA). Cleaned PCR products were sequenced using an ABI 3730 XL Sequencer (Xcleris Labs Pvt. Ltd., Ahmedabad, India).and the BigDye Terminator cycle sequencing kit (Applied Biosystems, Foster City, CA, USA). Products were analyzed directly on a 3730 XL DNA sequencer (Applied Biosystems). Both DNA strands were sequenced with primers ITS1 and ITS4 in separate reactions. PCR for amplification of *GAPDH* was performed as described for ITS above except that the annealing temperature used was 52 °C and the primers were GPD-1 and GPD-2 (12). All sequences were submitted to GenBank (Table 1) and subjected to the Basic Local Alignment Search Tool (BLAST) available at http://blast.ncbi.nlm.nih.gov/Blast.cgi (accessed on 12 October 2021).

### 4.3. Phylogenetic Analysis

ITS sequences for 20 isolates under study plus additional sequences of reference strains obtained from GenBank (Appendix A) were aligned using MEGA X [56] and then manually adjusted if needed in Mesquite [57]. Ends of the alignment were cut to make the analysis on common regions for all the taxa. Phylogeny trees derived from ITS sequences were constructed using maximum likelihood with substitution model K2 + g obtained by MEGA. Support for the branches was obtained with bootstrap, 500 replicates. Phylogeny trees were also obtained with Mrbayes (3.2.7a) [58]. The Bayesian analysis used the DNA substitution model of Kimara 2 parameter (K2 + G, nst = 2) with gamma distribution determined using MEGA X. Four chains and 1,000,000 Markov chain–Monte Carlo generations were run, and the current tree was saved to a file every 1000 generations. The stability of likelihood scores was confirmed with the plot of likelihood score versus generation number in Microsoft Excel. 25% of the initial trees were discarded as the burn–in phase. Posterior probabilities of above 95 were considered significant support for the clades. The maximum likelihood and the Bayesian trees were rooted to *Pyrenophora chaetomioides*.

*GADPH* trees of the twelve GenBank accessions (Appendix A) and the 19 isolates under investigation were also obtained with two methods, maximum likelihood with MEGA X and Bayesian, as described for ITS. The models used for both trees were K2 + G, obtained with MEGA X. Character status of the data was obtained with MEGA X.

### 4.4. Cultural and Morphological Variation

Single-spore-purified fungal cultures from maize leaf spots were maintained on PDA in 100 mm × 15 mm sterile polystyrene Petri plates (Fisher Scientific, Thane, India). For observations of morphological variability, 5 mm plugs of the seven-day-old culture of the isolates were placed in the center of the PDA plates and incubated at 27 ± 1 °C in a BOD incubator with alternate light and dark for 12 h daily. Observations were recorded in triplicate. Morphometric variations in the size of conidia and number of septa were examined at 100× magnification with the BX-53 microscope fitted with a camera and imaging software. The length and width of conidia as well as the number of septa were observed microscopically and compared with the identification key of Manamgoda et al., 2014 [4]; measurements being taken using Biovis Image Plus Software with advanced image analysis and image processing tools (developed by Expert Vision Labs, Mumbai, India). Averages of 10 conidia in a microscopic field are presented in Table 2. The observations on colony color and texture were recorded for 10 days after inoculation (DAI) from the top and bottom sides of the culture plates. The isolates were designated to different groups based on cultural characteristics. Observations on radial growth patterns were recorded at 24 h intervals and the final growth at 10 DAI is presented in Table 2. Average radial growth was recorded, and the cultures were assigned as fast (+++), medium (++), and slow (+) groups (Table 2).

### 4.5. Pathogenic Variability

Large-scale growth of inoculum was done on sorghum grains [59]. Sorghum grains were thoroughly washed in tap water, sterilized in 2 % sodium hypochlorite for 5 min, and soaked overnight in distilled water. Excess water from the soaked grains was drained off through several layers of cheesecloth, the grains dispensed into aliquots of 100 g each in 250 mL Erlenmeyer flasks, and autoclaved at 120 °C for 40 min. The sorghum grains were inoculated with bits (5 mm diameter) of actively growing culture of each of the fungal isolates. Cultures were shaken after every two days and incubated for 10 to 15 days at 28 °C. Colonized sorghum grains were dried under shade for 7 to 10 days and powdered. Simultaneously, seeds were planted in 20 inch-diameter pots in a sterilized soil mixture containing vermiculite, coco peat, and sand (2:2:1) in four replications of six pots each.

The Koch’s Postulates test [60] for pathogenicity of the 20 fungal isolates was conducted on the susceptible maize inbred line DHM117 in the greenhouse under controlled conditions (28 ± 2 °C temperature, relative humidity 85%, and 16 h photoperiod). Inoculation of maize seedlings was done using powdered sorghum grains. Twelve seedlings in six pots were inoculated twice *viz*., at the seven–leaf stage and eight days after the first inoculation, with a small pinch of inoculum (about 100 mg) applied in the leaf whorl as per standard techniques for disease resistance screening [61] (Appendix A). Twelve control plants were treated with sterilized healthy sorghum seed powder only. Pots were covered with polythene bags to maintain the desired humidity. The plants were observed daily, and disease scoring was done up to 20 days after inoculation. The inoculation experiment was performed a total of three times. Disease scoring was done using the rating scale of Payak and Sharma [58], which is based upon the severity of the infected leaves after 20 days of inoculation (1 to 5 rating scale) (Appendix A).

Very mild infection, as 1 to 2 or more scattered lesions on lower leaves of the host.Moderate infection showing few lesions on lower leaves only of the host.Moderate infection, with abundant lesions on lower leaves, spreading up to middle leaves and extending to upper leaves of the host.Severe infection showing abundant lesions on lower and middle leaves, extending to upper leaves of the host.Intense severity with abundant lesions on almost all the leaves showing premature drying or necrosis of infected leaf tissue.

Pathogens from the diseased leaf spots following inoculations were reisolated and observed to have similar morphology of the respective fungal inoculum.

### 4.6. Statistical Analysis

The data from cultural, morphological, and pathogenic variability was analyzed statistically to derive significance by SAS Ver 10.0. (SAS Institute, Cary, NC, USA), with desired statistic estimates such as Means, Standard Error (SE), Standard Deviation (SD), and Coefficient of Variation (CV). Percent Disease Index (PDI) was calculated based on an average of 10 replications using the formula:PDI = (Sum of all the ratings/maximum disease rating) × 100.

PDI > 70–100%—highly virulent (HV); PDI > 50–69%—Moderate virulence (MoV); PDI > 20–49%—Mild virulence (MV).

## Figures and Tables

**Figure 1 pathogens-10-01621-f001:**
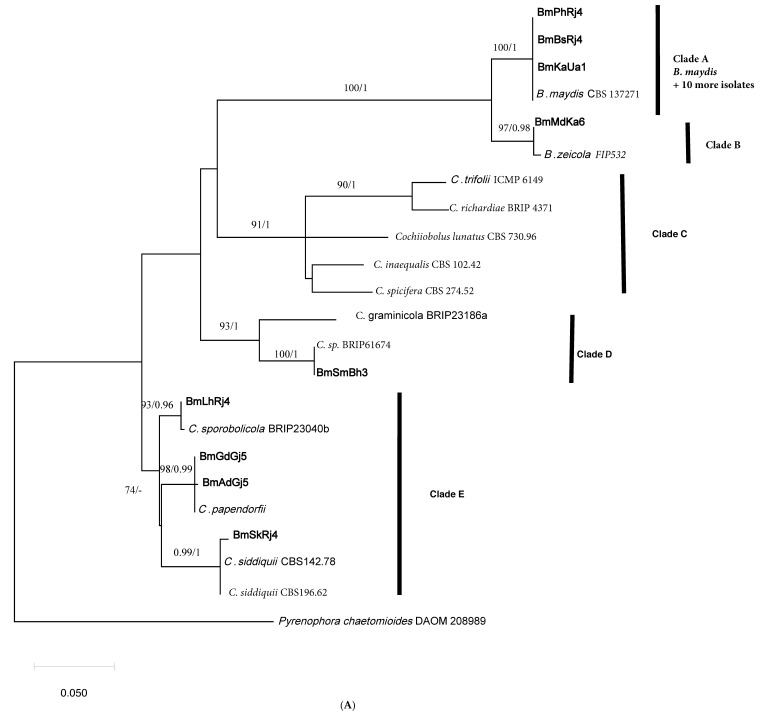
(**A**). Maximum likelihood tree obtained by MEGA X derived from glyceraldehyde 3-phosphate dehydrogenase (*GAPDH*) sequence data of 20 fungal isolates causing maize leaf blight (MLB) within reference sequences obtained from GenBank. The bootstrap values ≥ 70% and posterior probabilities ≥ 0.95 from Bayesian analysis are indicated above the branches, respectively. The scale bar refers to the number of nucleotide substitutions per site. The tree is rooted to *Pyrenophora chaetomiodes.* The leaf names in bold letters refer to isolates used in this investigation. (**B**). Maximum likelihood tree obtained by MEGA X derived from internal transcribed spacer (ITS) sequence data of 20 fungal isolates causing maize leaf blight (MLB) within reference sequences obtained from GenBank. The bootstrap values ≥ 70% and posterior probabilities ≥ 0.95 from Bayesian analysis are indicated above the branches, respectively. The scale bar refers to number of nucleotide substitutions per site. The tree is rooted to *Pyrenophora chaetomiodes*. The leaf names in bold letters refer to isolates used in this investigation.

**Figure 2 pathogens-10-01621-f002:**
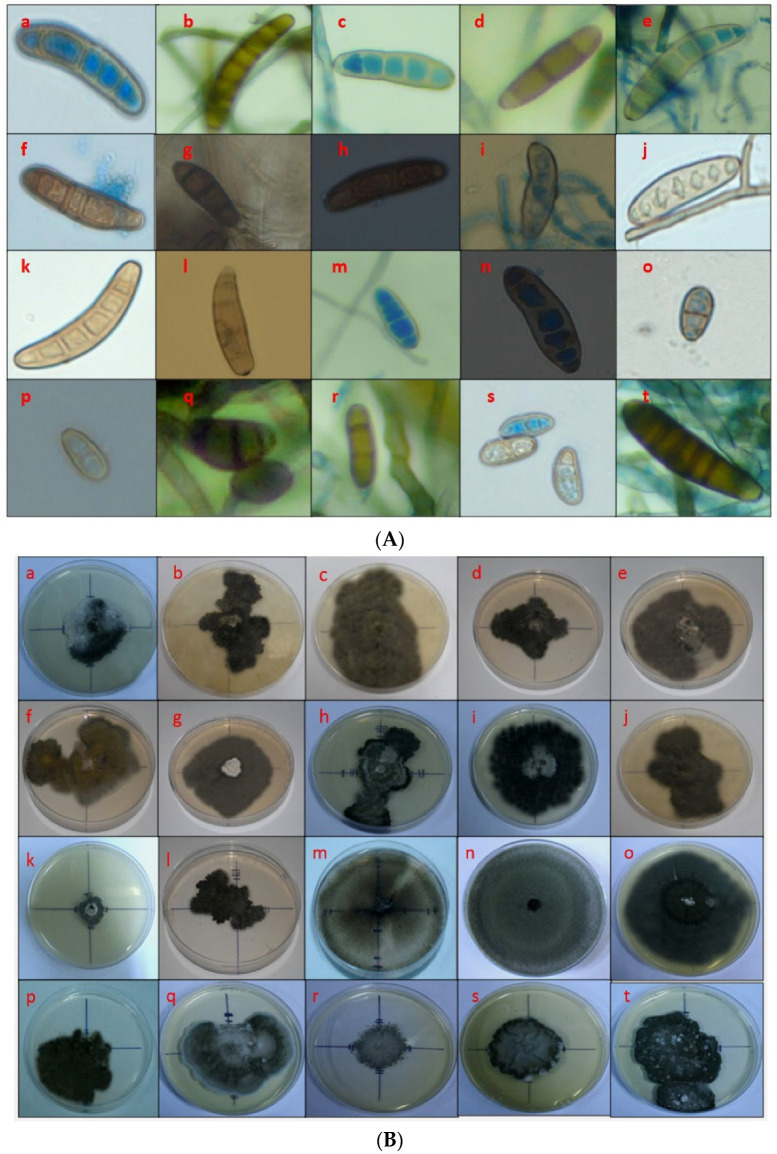
(**A**). Conidial morphology of various fungi. Panels a to m, culture of *Bipolaris maydis* isolates (Bar = 20 µm) (*BmPhRj4*, *BmBjUa1*, *BmBsRj4*, *BmDhBh3*, *BmCgRj4*, *BmPnDl2*, *BmDnRj4*, *BmKgUa1*, *BmKrHr2*, *BmKtRj4*, *BmLdPj2*, *BmPtUa1*, and *BmMyKa6*); panel n, *B. zeicola* (*BmMdKa6*); panels o & p, *Curvularia papendorfii* (*BmGdGj5* and *BmAdGj5*); panel q, *C. sporobolicola* (*BmLhRj4*); panel r, *C. siddiquii* (*BmSkRj4*); panel s, *C. graminicola*–like (*BmSmBh3*); panel t, *Alternaria* sp. (*BmAmRj4*). (**B**). Cultural variations among various fungi. Panels a to m, culture of *Bipolaris maydis* isolates *(BmPhRj4*, *BmBjUa1*, *BmBsRj4*, *BmDhBh3*, *BmCgRj4*, *BmPnDl2*, *BmDnRj4*, *BmKgUa1*, *BmKrHr2*, *BmKtRj4*, *BmLdPj2*, *BmPtUa1*, and *BmMyKa6*); panel n, *B. zeicola* (*BmMdKa6*); panels o & p, *Curvularia papendorfii* (*BmGdGj5 and BmAdGj5*); panel q, *C. sporobolicola* (*BmLhRj4*); panel r, *C. siddiquii* (*BmSkRj4*); panel s, *C. graminicola*–like (*BmSmBh3*); panel t, *Alternaria* sp. (*BmAmRj4)*. (**C**). Symptoms of maize leaf blight (MLB) caused by various fungi. Panels a to m, symptoms caused by *Bipolaris maydis* isolates (*BmPhRj4*, *BmBjUa1*, *BmBsRj4*, *BmDhBh3*, *BmCgRj4*, *BmPnDl2*, *BmDnRj4*, *BmKgUa1*, *BmKrHr2*, *BmKtRj4*, *BmLdPj2*, *BmPtUa1*, and *BmMyKa6*); panel n, *B. zeicola* (*BmMdKa6*); panels o & p, *Curvularia papendorfii* (*BmGdGj5* and *BmAdGj5*); panel q, *C. sporobolicola* (*BmLhRj4*); panel r, *C. siddiquii* (*BmSkRj4*); panel s, *C. graminicola*–like (*BmSmBh3*); panel t, *Alternaria* sp. (*BmAmRj4*); u & v are mock–inoculated control and uninoculated samples, respectively.

**Table 1 pathogens-10-01621-t001:** Details of maize leaf blight causing isolates from *Zea mays* collected from different agro-ecological zones under maize production.

S. No. ^$^	New Code	Location	AEZ *	State	Latitude	Longitude	Soil Type	No. of Samples Collected	Accession No.	
									ITS	*GAPDH*
1	*BmPhRj4*	Pichola (Udaipur)	WDR	Rajasthan	24°36′48.00″ N	73°40′48.00″ E	Sandy/clay loam to dessert loam	15	KX668613	OL519604
2	*BmBjUa1*	Bajaura	WHR	Uttaranchal	31°50′54.2483″ N	77°9′51.6013″ E	Sandy to clay loam	23	KX668605	OL519605
3	*BmBsRj4*	Banswara (Udaipur)	WDR	Rajasthan	23°32′48.3252″ N	74°26′1.7880″ E	Sandy/clay loam to dessert loam	11	KX668614	OL519606
4	*BmDhBh3*	Dholi	MGP	Bihar	25°51′25.9951″ N	85°46′85.5895″ E	Deep loamy/silt/clay loam	27	KX668619	OL519607
5	*BmCgRj4*	Chittorgarh (Udaipur)	WDR	Rajasthan	24°54′16.5716″ N	74°42′29.558″ E	Sandy/clay loam to dessert loam	18	KX668617	OL519608
6	*BmPnDl2*	Patel Nagar	TGP	Delhi	28°39′8.7966″ N	77°11′29.9389″ E	Alluvium	15	KX668610	OL519609
7	*BmDnRj4*	Dungarpur (Udaipur)	WDR	Rajasthan	23°50′16.5716″ N	73°50′29.558″ E	Sandy/clay loam to dessert loam	19	KX668616	OL519610
8	*BmKgUa1*	Kangra	WHR	Uttaranchal	32°5′59.2944″ N	76°16′8.7744″ E	Shallow to deep loam	21	KX668609	OL519611
9	*BmKrHr2*	Karnal	TGP	Haryana	29°41′8.4944″ N	76°59′25.737″ E	Sandy clay	20	KX668608	OL519612
10	*BmKtRj4*	Kota (Udaipur)	WDR	Rajasthan	24°10′16.5716″ N	75°52′29.558″ E	Sandy/clay loam to dessert loam	23	KX668618	OL519614
11	*BmLdPj2*	Ludhiana	TGP	Punjab	30°54′3.474″ N	75°51′26.1929″ E	Deep loamy/sandy/clay loam	34	KX668623	OL519613
12	*BmPtUa1*	Pantnagar	WHR	Uttaranchal	29°1′15.74″ N	79°29′23.06″ E	Sandy clay	18	KX668604	OL519615
13	*BmMyKa6*	Mysore	SPHR	Karnataka	12°31′25.4316″ N	76°53′40.8624″ E	Light red sandy loam	9	OK576634	OL519616
14	*BmMdKa6*	Mandya	SPHR	Karnataka	12°31′25.4316″ N	76°53′40.8624″ E	Light red sandy loam	13	KX668606	OL502169
15	*BmGdGj5*	Godhra	GPHR	Gujarat	22°46′24.9456″ N	73°36′49.9824″ E	Sandy loam	12	KX668621	OL502170
16	*BmAdGj5*	Anand	GPHR	Gujarat	22°33′14.5044″ N	72°56′56.1696″ E	Sandy loam	12	KX668622	OL502171
17	*BmLhRj4*	Lakhawali (Udaipur)	WDR	Rajasthan	24°34′16.5716″ N	73°41′29.558″ E	Sandy/clay loam to dessert loam	11	KX668612	OL502172
18	*BmSkRj4*	Sukher (Udaipur)	WDR	Rajasthan	24°34′16.5716″ N	73°41′29.558″ E	Sandy/clay loam to dessert loam	16	KX668615	OL519603
19	*BmSmBh3*	Samastipur	MGP	Bihar	25°51′46.6848″ N	85°46′51.7044″ E	Deep loamy/silt/clay loam	18	KX668620	OL519617
20	*BmAmRj4*	Amberi (Udaipur)	WDR	Rajasthan	26°55′16.5716″ N	73°50′29.558″ E	Sandy/clay loam to dessert loam	15	KX668611	**

* AEZ = Agro-ecological zone; WHR = Western Himalayan Region; TGP = Transgangetic Plains; MGP = Middle Gangetic Plains; SPHR = Southern Plateau and Hill Region; WDR = Western Dry Region. ** *GAPDH* sequencing not done. **^$^** Based on phylogeny, serial numbers (SN) 1 to 13 were identified as *Bipolaris maydis*, SN 14 as *B. zeicola*, SN 15 and 16 as *Curvularia papendorfii*, SN 17 as *C. sporobolicola*, SN 18 as *C. siddiquii*, SN 19 as *Curvularia* sp. (*C. graminicola*-like), and SN 20 as *Alternaria* sp.

**Table 2 pathogens-10-01621-t002:** Cultural characteristics and pathogenic profiles of maize leaf blight (MLB)-causing isolates from *Zea mays* collected from different agro-ecological zones under maize production on PDA at 27 ± 1 °C.

S N ^$^	Isolate	Size of Conidia (µm) and Septations *			Radial Growth (in mm) **	Disease Score ***	Incubation Period (in h) ^∞^	Colony Texture ^#^	Colour of Colony **	Disease Index ^β^ (PDI)
		Length (µm)	Width (µm)	No. of Septa						
1	*BmPhRj4*	9.8 (6.54–17.86)	3.4 (2.25–4.63)	2.7 (2–5)	47.5 (++)	1.5	72 (Type-I)	*Rr/Nz/Irm*	Light black	30 (MV)
2	*BmBjUa1*	16.9 (10.68–20.84)	4.1 (3.72–4.45)	7.1 (5–8)	58.5 (++)	2.6	72 (Type-II)	*Rr/Nz/Rm*	Black	52 (MoV)
3	*BmBsRj4*	12.7 (7.79–22.78)	4.9 (4.36–5.26)	4.7 (3–6)	71.4 (+++)	2.3	48 (Type-III)	*Sap/Nz/Rm*	Black	46 (MV)
4	*BmDhBh3*	14.4 (11.91–16.66)	4.5 (3.91–5.43)	3.7 (3–5)	51.1 (++)	3.7	72 (Type-I)	*Rap/Nz/Rm*	Black	74 (HV)
5	*BmCgRj4*	14.8 (9.26–20.96)	4.3 (2.48–4.78)	4.2 (2–9)	72.3 (+++)	2.9	48 (Type-II)	*Rr/Nz/Rm*	Light black	58 (MoV)
6	*BmPnDl2*	19.3 (13.27–21.36)	5.0 (3.91–5.73)	4.3 (3–7)	68.8 (++)	2.7	72 (Type-III)	*Rr/Z/Irm*	Black	54 (MoV)
7	*BmDnRj4*	11.1 (6.52–14.94)	4.7 (3.63–5.51)	3.0 (2–5)	69.0 (++)	1.9	48 (Type-II)	*Sap/Nz/Rm*	Light grey	38 (MV)
8	*BmKgUa1*	10.1 (7.34–15.43)	4.5 (3.89–5.46)	3.1 (2–6)	63.4 (++)	2.4	76 (Type-II)	*Sap/Nz/Rm*	Dark grey	48 (MV)
9	*BmKrHr2*	15.1 (19.25–11.03)	4.4 (4.19–4.93)	6.1 (4–7)	74.3 (+++)	3.5	48 (Type-I)	*Sap/Nz/Rm*	Dark grey	70 (HV)
10	*BmKtRj4*	13.0 (8.92–19.4)	3.9 (2.54–5.22)	3.7 (2–5)	54.3 (++)	1.7	96 (Type-III)	*Sap/Nz/Rm*	Black	34 (MV)
11	*BmLdPj2*	14.3 (8.78–23.57)	4.4 (2.55–5.77)	5.1 (3–6)	37.4 (+)	4.3	72 (Type-II)	*Rr/Nz/Irm*	Dark grey	86 (HV)
12	*BmPtUa1*	14.7 (11.89–17.74)	4.9 (4.23–5.43)	4.1 (3–4)	45.5 (++)	3.2	72 (Type-II)	*Rr/Nz/Rm*	Dark grey	64 (MoV)
13	*BmMyKa6*	13.1 (10.77–17.03)	3.8 (2.46–4.78)	4.1 (3–4)	78.9 (+++)	2.7	72 (Type-IV)	*Sap/Z/Rm*	Light grey	54 (MoV)
14	*BmMdKa6*	11.6 (6.88–15.32)	3.6 (4.12–5.36)	3.8 (4–2)	81.8 (+++)	1.7	72 (Type-I)	*Sap/Z/Rm*	Light grey	34 (MV)
15	*BmGdGj5*	7.6 (6.83–9.77)	3.1 (3.61–4.52)	1.0 (1–3)	79.3 (+++)	3.3	48 (Type-I)	*Rr/Nz/Rm*	Black	66 (MoV)
16	*BmAdGj5*	12.1 (8.23–15.98)	4.3 (3.84–5.86)	3.0 (2–9)	51.6 (++)	4.5	72 (Type-VI)	*Rap/Nz/Rm*	Black	90 (HV)
17	*BmLhRj4*	17.2 (10.33–23.13)	3.7 (2.44–4.57)	5.0 (4–8)	59.3 (++)	3.7	72 (Type-III)	*Rr/Z/Irm*	Dark grey	74 (HV)
18	*BmSkRj4*	16.8 (9.78–20.11)	3.1 (2.98–4.91)	3.0 (2–8)	37.7 (+)	3.6	72 (Type-V)	*Sap/Nz/Rm*	Dark grey	72 (HV)
19	*BmSmBh3*	7.8 (6.12–10.11)	6.1 (4.21–7.93)	2.0 (2–6)	42.6 (++)	4.3	76 (Type-V)	*Sap/Z/Rm*	Dark grey	86 (HV)
20	*BmAmRj4*	18.6 (7.33–21.67)	5.8 (3.22–8.44)	6.0 (4–8)	67.3 (++)	4.1	72 (Type-IV)	*Rr/Nz/Irm*	Black	82 (HV)
CD at 5%		6.2	1.2	2.0	1.28	1.1				

* Range is given in parentheses. ** Radial growth was recorded 10 days after inoculation (DAI) and expressed as an average of 3 replications; (+) = Slow growth (30–40 mm), (++) = Medium growth (40–70 mm), (+++) = Fast growth (70–90 mm). ***^#^***
*Rr* = rough raised, *Z* = Zonation, *Nz* = no zonation, *Rm* = regular margin, *Rap* = rough appressed, *Irm* = irregular margin; *Sap* = smooth appressed, *Sr* = smooth raised. *** Disease score expressed as an average of 10 replications using a rating scale of 1–5 (Payak and Sharma, 1983). ^β^ PDI (Percent Disease Index) expressed as an average of 10 replications using the formula PDI = sum of all ratings/maximum disease rating × 100; PDI > 70–100%—highly virulent (HV), PDI > 50–69%—Moderate virulence (MoV), PDI > 20–49%—Mild virulence (MV). **^∞^** Symptoms: Type-I, Small, dot–like yellowish necrotic lesions scattered away from midrib; Type-II, Dot–like, tan–colored lesions scattered profusely on the leaf surface; Type-III, Elongated, nearly long strip, tan–colored lesions restricted by veins; Type-IV, Long, narrow tan–colored linear lesions; Type-V: Circular lesions that were larger than the Type I lesions with purplish margins; Type-VI: Very large necrotic lesions along the length of leaf margins or parallel to midrib. **^$^** Based on phylogeny, serial numbers (SN) 1 to 13 were identified as *Bipolaris maydis*, SN 14 as *B. zeicola*, SN 15 and 16 as *Curvularia*
*papendorfiii*, SN 17 as *C. sporobolicola*, SN 18 as *C. siddiquii*, SN 19 as *Curvularia* sp. (*C. graminicola*–like), and SN 20 as *Alternaria* sp.

## Data Availability

The datasets generated during and/or analyzed during the current study can be find in the main text and the Appendix A.

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
