# Peer review of "Fungal Species Causing Maize Leaf Blight in Different Agro-Ecologies in India"

_pathogens, 2021, doi:10.3390/pathogens10121621_

Round 1

Reviewer 1 Report

In the submitted manuscript by Singh et al., entitled “Fungal Species Causing Maize Leaf Blight in Different Agro-ecologies in India ” the authors isolated and characterized the fungal genera causing Maize Leaf Blight disease across different parts of India. Based on morphology, pathogenicity, and phylogenetic analysis of 20 representative isolates they confirmed that the majority of isolates were Bipolaris maydis and others include different Curvularia species, one B. zeicola, and one Alternaria sp. Moreover, the authors reinfected and confirmed their pathogenicity on Maize leaves.

Overall, the experiments included in this manuscript are informative and generally, the interpretations seem reasonable. This is a valuable study and the data are well presented. The manuscript writing is understandable and fairly easy to read. In general, this manuscript presents convincing evidence about the fungal diversity causing MLB disease in India. However, as I detail below, there are several issues including missing information, data, and controls in figures that should be addressed.

Line 103: “one candidate isolate was selected from each of the 20 hotspots,” it is unclear why only one isolate was selected from each hotspot? Were all the isolates from one hotspot had similar morphological characteristics? Whether samples were collected from one field or different fields at one hotspot? Please clarify or discuss.

Line 107: “Isolates were examined for typical characteristics of B. maydis”. Why do authors only look for characteristics of B. maydis? In this case, they might have missed other fungal species.  

Line 159, 160, 169, and 176: referencing Fig.4 and Fig. 5 should instead reference fig. 1A and 1B.

Line 185-187, Line 219-223: Either write the name of the isolates indicated in the table or include the location in the table. Otherwise, it is difficult to correlate with the corresponding text.

Fig. 2A: Include scale in the microscopic picture.

Fig. 2B: In the figure legend, include names of the isolates instead of serial no. for clarity and for better correlation with the corresponding text in the results section.

Fig. 2C: Not clear about the replication of this experiment. How many technical or biological replicates? Show picture for mock-treated control. Please mention the name of the isolates in the fig. legend.

Reviewer 2 Report

Dear Authors, although the subject of the work seems important and the goal is significant for the proper management of maize fungal diseases in India. First of all, the isolation of 20 fungal strains suspected of causing MLB from 350 samples collected from various agro-ecological zones in India is unrepresentative to conclude on morpho-patho-genetic variation. Therefore, the conclusions are not substantiated.  In addition, the methodology is incompletely described, not all results have been documented, and the text contains many incorrect / erroneous descriptions of genes, studied regions of DNA, etc. Also, the italics were not used for names of eg species or genera. In general, the entire manuscript requires major linguistic revision and formatting according to the style of the journal. Detailed directions, comments and notes are marked in the PDF version of the manuscript. 

Round 2

Reviewer 2 Report

Dear Authors, the current version of the manuscript is suitable for publication in Pathogens.